# Change Patterns in Corneal Intrinsic Aberrations and Nerve Density after Cataract Surgery in Patients with Dry Eye Disease

**DOI:** 10.3390/jcm11195697

**Published:** 2022-09-27

**Authors:** Dalan Jing, Xiaodan Jiang, Xiaotong Ren, Jie Su, Shanshan Wei, Ran Hao, Yilin Chou, Xuemin Li

**Affiliations:** 1Beijing Key Laboratory of Restoration of Damaged Ocular Nerve, Department of Ophthalmology, Peking University Third Hospital, 49 North Garden Rd., Haidian District, Beijing 100191, China; 2Beijing Ophthalmology & Visual Sciences Key Laboratory, Beijing Tongren Eye Center, Beijing Tongren Hospital, Capital Medical University, Beijing 100069, China; 3Department of Ophthalmology, BenQ Medical Centre, The Affiliated BenQ Hospital of Nanjing Medical University, Nanjing 210017, China

**Keywords:** dry eye disease, corneal nerve, corneal vortex, corneal intrinsic aberrations, cataract surgery

## Abstract

This study aimed to evaluate the change patterns in corneal intrinsic aberrations and nerve density after cataract surgery in dry eye disease. The preoperative, 1- and 3-month postoperative dry eye-related parameters were obtained by the Oculus keratograph and the ocular surface disease index questionnaire. The corneal intrinsic aberrations were measured using the Pentacam HR system. In vivo confocal microscopy was performed to observe the vortical and peripheral corneal nerves. An artificial intelligence technique run by the deep learning model generated the corneal nerve parameters. Corneal aberrations on the anterior and total corneal surfaces were significantly increased at 1 month compared with the baseline (*p* < 0.05) but gradually returned to the baseline by 3 months (*p* > 0.05). However, the change in posterior corneal aberration lasted up to 3 months (*p* < 0.05). There was a significant decrease in the corneal vortical nerve maximum length and average density after the operation (*p* < 0.05), and this damage lasted approximately 3 months. The corneal vortical nerve maximum length and average density were negatively correlated with the anterior corneal surface aberrations before and 1 month after the operation (correlation coefficients, CC = −0.26, −0.25, −0.28; all *p* < 0.05). Corneal vortex provided a unique site to observe long-term corneal nerve injury related to eye dryness. The continuous damage to the corneal vortical nerve may be due to the continuous dry eye state.

## 1. Introduction

Modern cataract surgery has reached higher standards of safety and visual performance. While many patients have complained of postoperative dry eye symptoms, studies have documented an increase in the prevalence and severity of dry eye diseases (DEDs) [1,2]. There are numerous causes of aggravated eye dryness, such as damage to the corneal nerve and epithelium caused by surgical incisions [3], light exposure from the operating microscope [4], topical anesthesia and eye drops containing preservatives [5]. One study reported a reduction in the corneal nerve length and density after cataract surgery [6]. Previous studies reported that the corneal subbasal nerve density and length tended to decrease in DED, indicating damaged nerve fibers [7]. They indicated that cataract surgery aggravated the eye dryness by damaging the corneal nerves. These changes may be attributed to inflammation—a mild and persistent inflammatory state that leads to the recruitment and maturation of dendritic cells due to the release of inflammatory mediators—prostaglandins, leukotrienes, cytokines and neurotrophic factors from the damaged corneal tissue and inflammatory cells. These factors stimulate epithelial growth, proliferation and differentiation [3,8,9]. The ensuing corneal nerve dysfunction contributes to dry eye progression, ocular pain and discomfort, and other neuropathic symptoms [10]. Their study location focused on the center of the cornea, while due to the less observed range and on positioning function, each measurement was conducted in a different position, notably increased variability, poor reproducibility and decreased comparability. An effective and standard examination location is needed for the preoperative and postoperative comparison. The corneal vortex is a recently recognized peculiar structure that is universally present in the population [11]. Additionally, it is easy to recognize and has a relatively fixed site compared with the corneal center [12].

Guirao et al. [13] found increased corneal aberrations after conventional cataract surgery. In particular, the development of higher-order aberrations (HOAs) can affect visual quality [14]. Patients with DED have greater optical aberrations, mainly in the cornea. This can be attributed to the instability and irregularity of the tear film. These symptoms naturally lead to visual function impairment, thus affecting one’s quality of life [15,16]. However, these studies did not pay attention to corneal intrinsic aberrations. There are many similarities in ocular surface changes between the two. Prior studies have focused on one specific aspect: changes in corneal nerves or aberrations.

Therefore, in our study, we investigated the changes in corneal intrinsic aberrations, corneal regularity, corneal nerve density, corneal endothelial number and symptoms and signs of eye dryness in DED patients before and after cataract operation using a comprehensive approach that combines the ocular signs, symptoms, accurate equipment—such as Pentacam—in vivo confocal microscopy (IVCM) and Oculus keratography. Moreover, these parameters were further evaluated to identify their relationship.

## 2. Materials and Methods

### 2.1. Study Design and Population

This is a prospective study conducted at the Eye Center of Peking University Third Hospital, China. A total of 106 eyes of 106 patients with eye dryness meeting the inclusion and exclusion criteria were evaluated before and after cataract surgery. Participants were eligible if the patients were diagnosed with age-related cataracts and DED based on the 2017 Report of the Tear Film & Ocular Surface Society International Dry Eye Workshop (TFOS DEWSII) [17]. The criteria for the diagnosis of eye dryness included symptoms of dryness, foreign body sensation, burning, fatigue or blurred vision, and tear break-up time (TBUT) < 10 s. The exclusion criteria were as follows: age younger than 50 years or older than 80 years; patients who currently received DED treatment (other than artificial tears), such as punctual plugs, thermal treatment; any ocular surface and corneal abnormalities (including inflammation, trauma, degeneration et al.); history of contact lens wear within 1 month; recent eye surgery; nasolacrimal duct obstruction; glaucoma; ocular fundus diseases; diabetes and immunologic systemic disease; regular corneal astigmatism ≥ 0.75 D or irregular astigmatism.

The right eye was routinely enrolled. If the right eye did not meet the inclusion criteria, the left eye was enrolled instead.

### 2.2. Ethic

This study adhered to the tenets of the Declaration of Helsinki and was approved by the Ethics Committee of the Peking University Third Hospital (#M2019236). All patients were fully informed of the details and possible risks of the procedure, and written informed consent was obtained from all participating patients.

### 2.3. Procedure

Preoperative clinical assessment (baseline assessment) included the following: collection of demographic information, OSDI questionnaire [18], Catquest questionnaire [19], best corrected Snellen visual acuity (BCVA) (converted to logarithm of the minimum angle of resolution or (logMAR) for the purpose of statistical analysis), and slit-lamp biomicroscopy. The keratograph 5 M (Oculus Optikgeräte GmbH, Wetzlar, Germany) was used to measure the noninvasive tear break-up time (TBUT), tear meniscus height (TMH) and meibography scores [20,21]. The Pentacam HR system (Oculus Inc., Wetzlar, Germany) was used to measure corneal intrinsic aberrations and corneal surface regularity indices. IVCM was performed to assess subbasal plexus morphology and endothelial cell counts. An interval of 5 min was required between the different examinations. To avoid bias, each examination was performed by the same experienced clinician (J.D.L. and J.X.D.).

The Pentacam HR system is a noninvasive system for measuring and featuring the anterior segment. The raw data from the Scheimpflug camera were obtained in the 50-picture corneal-fine scan mode over a measurement zone diameter of 4.0 mm on the anterior corneal surface and posterior corneal surface. The elevation data of the rotating Scheimpflug camera images were reconstructed to a 3-dimensional corneal structure, which can eliminate the tear film’s effect, thus measuring corneal intrinsic aberrations. This allows the quantification of corneal surface regularity as computed indices, namely the index of surface variance (ISV), index of vertical asymmetry (IVA), index of height asymmetry (IHA) and index of height decentration (IHD). The Zernike polynomials of the corneal wavefront were automatically generated using the integrated software. The Zernike coefficients up to eighth order over the central 4.0 mm diameter zones of total, anterior and posterior corneal surfaces were extracted, including the total root mean square (RMS) aberrations, RMS of the higher-order aberrations (RMS HOA) and RMS of the lower-order aberrations (RMS LOA).

Corneal images were obtained by the IVCM and had a definition of 384 × 384 pixels over an area of 400 µm × 400 µm, with a lateral spatial resolution of 0.5 µm and a depth resolution of 1–2 µm. ICVM was performed on each eye in two different areas: corneal vortex and superior peripheral cornea. The periphery was determined as an outside one-sixth part of the cornea (apart from the temporal surgical incision). Approximately 30 pictures were captured from the corneal epithelium to the endothelium, and images with good quality were selected by the same experienced masked ophthalmologist for analysis.

We selected each of the five high-quality images in the corneal vortex and periphery with no overlapping for the analysis of the corneal nerve parameters by CNS-Net. The method demonstrated high accuracy, fast speed, and was quantified in subbasal corneal nerve segmentation with IVCM [22]. The AI technique produced the maximum length of the corneal vortical nerve, the average density of the corneal vortical nerve, the maximum length of the corneal peripheral nerve and the average density of the corneal peripheral nerve. As regards the corneal endothelium count, we selected a clear image for cell counting, and the counting was completed by an automatic camera.

All examinations were performed at baseline, after 1 month and after 3 months. All patients were given levofloxacin 0.5% eye drops (Santen, Osaka, Japan) four times a day from three days before surgery. Povidone–iodine (PI) 5% solution was instilled before and after the placement of the lid speculum. The patients underwent standard phacoemulsification through a 3.0 mm limbal incision at the 12 o’clock position, followed by a paracentesis incision at the 3 o’clock position, and intraocular lens implantation (Tecnis ZCB00, Abbott, Abbott Park, IL, USA) by the same surgeon (L.X.M.). The Centurion microsurgical system (Alcon Laboratories, Inc., Ft. Worth, TX, USA) was employed for phacoemulsification and cataract removal. Prednisolone acetate 1% eye drops (Allergan, Ireland) and levofloxacin 0.5% eye drops were used four times per day for one week and were then gradually reduced in both groups.

### 2.4. Statistical Analysis

We used GPower 3.1 to calculate the sample size. The effect size, significance (alpha) and statistical power (1-beta) were set to 0.3, 0.05 and 0.95, respectively, which generated an estimated total sample size of 104 to achieve statistical significance. All analyses were performed using SPSS version 23.0 software (IBM Corp., Armonk, New York, NY, USA). We verified the normality of the data distribution by the Kolmogorov-Smirnov test. Descriptive parameters were expressed as the number of patients (%) or mean ± standard deviation (SD)/median with an interquartile range, depending on the distribution pattern. The paired t test and Wilcoxon signed rank test were used for the preoperative data and postoperative data comparisons. The Pearson and Spearman correlation coefficients were calculated. A *p* value less than 0.05 was considered statistically significant.

## 3. Results

One hundred and six eyes of 106 patients (mean age, 69.32 ± 8.91 years) were recruited. Thirty-nine participants (36.7%) were men. No eye had surgical complications throughout the 3-month follow-up.

Table 1 summarizes the comparison results of visual function, ocular symptoms and signs before and after cataract. The BCVA and Catquest-9SF scores in all cases were significantly better 1-month and 3-months after than they were baseline, while there were no significant differences in these scores between at 1-month and 3-months. The OSDI score was significantly decreased at 3 months compared to that at baseline. Ocular pain was elevated at 1-month compared with baseline. Poor vision at 1-month and 3- months was improved compared to that at baseline. However, other symptoms did not change significantly. After 1-month, TMH was significantly better than baseline. The TBUT and meibography scores worsened after 1 month compared with those at baseline.

The number of corneal endothelial cells significantly decreased after 1 month and 3 months. There was a significant decrease in the corneal vortical nerve maximum length and average density after the operation, and this damage was present after 3 months. However, for corneal peripheral nerves, this damage only lasted 1 month. Other comparisons among ICVM parameters are shown in Table 2.

Patients had increased anterior and total corneal HOAs at 1 month. Significant changes in the posterior corneal HOAs occurred after 1 month and 3 months. Total corneal aberration analysis found that there were significant changes at both timepoints; the rule is that aberration increases and then decreases. ISV and IVA increased at 1 month, however, the IVA returned to the baseline values at 3 months. Further details are provided in Table 3.

Correlation between ICVM parameters and corneal intrinsic aberrations before and after cataract operation were measured. The corneal endothelial number of 1 month was negatively correlated with RMS HOA (CF) (CC = −0.27; *p* < 0.05). However, after 3 months, it was positively correlated with RMS (CB) and RMS LOA (CB) (CC = 0.59, 0.59; all *p* < 0.01). The corneal vortical nerve maximum length and average density were negatively correlated with the total and anterior corneal surface aberrations before surgery (CC = −0.26, −0.25, −0.25, −0.26; all *p* < 0.05). At 1 month, only the corneal vortical nerve maximum length was negatively correlated with the anterior corneal surface aberrations (CC = −0.28; *p* < 0.05). At 3 months, the corneal vortical nerve maximum length and average density were positively correlated with posterior corneal surface aberrations (CC = 0.61, 0.57; all *p* < 0.01). The corneal peripheral nerve average density at baseline was negatively correlated with RMS LOA (cornea) and IHD (CC = −0.21, −0.22; all *p* < 0.05). At 3 months, it was negatively correlated with RMS HOA (CF) and RMS HOA (cornea) (CC = −0.43, −0.53; all *p* < 0.05).

## 4. Discussion

Ophthalmic surgery may have the potential to temporarily induce or aggravate dry eye conditions such as photorefractive keratectomy and laser-assisted in situ keratomileusis (LASIK), especially during the short postoperative stage [23,24]. Furthermore, cataract surgery has been shown to have the same effects on dry eye symptoms, with eye dryness being one of the most frequent complaints during the postoperative period [25]. The multifactorial etiology of eye dryness is still not fully understood [26]. Patients with DED who experience the stimulation of cataract surgery have increased disease severity. In the past, the academic study of postoperative eye dryness has focused on changes in ocular symptoms and signs, such as the OSDI questionnaire, TBUT, TMH, corneal fluorescein staining and Schirmer test. Combined, these series are commonly used for the assessment of the degree of severity. When symptoms and signs are inconsistent, these methods provide poor direct information on an ocular surface status. It is widely known whether DED or cataract surgery damages the corneal nerve by ICVM examination. Moreover, both can increase corneal aberration by Pentacam. To more comprehensively assess the ocular surface status, our study presents a continuous and comprehensive evaluation system for eye dryness during the perioperative cataract period by combining locatable IVCM, Pentacam and Oculus keratograph.

Interestingly, we found that the OSDI score at 3 months was significantly lower than it was at baseline. However, this does not mean that the dry eye symptoms will improve 3-months after surgery. Because visual acuity is significantly improved after cataract surgery, which affects the OSDI score, we compared each dry eye symptom of the OSDI questionnaire. Poor vision at 3 months was improved compared with that at baseline, whilst other symptoms did not change significantly. Three months after the operation, however, the dry eye symptoms did not improve and the dry eye status persisted. Choi et al. [27] showed that the TBUT was not notably altered during the perioperative periods after cataract operation. Consistently with this report, we also found these results. Some studies have shown a reduced TBUT only during the early postoperative phase [28,29]. Others reported that the TBUT was decreased after 1 month or 3 months [2,30]. These divergences could be partly attributed to the differences in the measuring method. A keratograph was used, which permits an automated, hypersensitive and examiner-independent technique for measuring TBUT. As measured using the Keratograph, TBUT was consistently less than it was in the subjective observer recordings, since the former can record the first incident of break-up anywhere in the tear film [31]. Another reason for these differences could be the heterogeneity of the enrolled population in each study. We only enrolled DED patients, whilst others enrolled patients with and without DED. Additionally, we evaluated the function of the meibomian gland throughout the trial, finding that the meibography scores dramatically deteriorated after 1 month. One study reported that meibomian gland function may be altered without accompanying structural changes after cataract surgery [30]. Cataract surgery might have caused meibomian gland obstruction. Our results suggest possible transient structural changes in the meibomian gland related to cataract surgery or medication that aggravated the symptoms of ocular surface discomfort. We assessed the lower tear meniscus using Oculus keratograph, which has shown high repeatability [32]. However, TMH showed a significant increase at 1-month. This result is very confusing, which was mixed with many factors. One reason may be attributed to the application of postoperative eye drops, which automatically improves TMH. Additionally, the painful eyes were aggravated at 1-month compared to baseline. This could explain why TMH was increased at 1 month, owing to irritant tear secretion.

Previous studies have shown that the injured nerves degenerate after cataract surgery, resulting in reduced subbasal nerve density, reflectivity and increased beading [3,6]. Previous studies have usually measured the central cornea by IVCM due to its positioning function but in spite of its lesser observation range. The measurement from which these results were obtained were conducted in different positions, with increased variability, poor reproducibility and decreased comparability. The corneal vortex is a recently recognized peculiar structure that is universally present in the population [11]. Moreover, it is easy to recognize and has a relatively fixed site that is located approximately 1–2 mm below or towards to the nasal from the corneal apex [33]. Patel et al. suggested that the mean subbasal nerve density was at its greatest centrally [34]. To assess the corneal nerve at different locations, we measured the corneal vortex and periphery separately before and after cataract. The corneal vortical nerve average density and maximum length were significantly reduced until after 3 months or longer. However, damage was observed around the corneal periphery only until 1 month. The corneal density and length have not yet returned to the preoperative status of the last phase. Consistent with early studies, the corneal central and temporal incised subbasal nerve density was reduced one month after cataract surgery in patients [3,6]. These changes are due to a temporal clear cornea incision to a small extent, as also occurs in the corneal center. Our research also confirmed this conjecture. However, we tested the corneal vortex, which is more fixed, and the corneal periphery apart from the temporal surgical incision. We ruled out the effect of corneal incision and found that there were other factors affecting the corneal nerve. Some people consider it to be due to ultrasonic energy damage, ocular medicated toxicity or others related to surgery. Interestingly, the damage to the peripheral corneal nerve lasted only 1 month, however, central corneal damage lasted until 3 months or more. It seems to have other factors that continue to harm the corneal vortex. However, there have been no relevant studies concerning the corneal vortical nerve after cataract surgery or eye dryness. Then, referring to the studies about the corneal center, we hypothesized that the persistent injury of the corneal vortical nerve was due to the lasting dry eye conditions, in other words, inflammation [35]. We will explain this in combination with the changes in corneal aberration.

Patients with eye dryness have greater corneal aberration, which is usually due to the instability of the tear film [36]. Based on our previous studies [37], DED patients had increased corneal intrinsic aberrations except for the aberration caused by tear film. Pentacam was used to measure corneal intrinsic aberrations. Our research found that the anterior, posterior and total corneal surface higher-order aberrations increased 1 month after the operation. There was no difference in corneal intrinsic aberrations after the operation and before the operation except for the posterior HOAs that increased at 3 months. They believed that corneal aberrations may be related to the postoperative ocular surface conditions, such as dry-eye symptoms. The current strategy for dry eye treatment is also mainly based on eliminating exacerbating factors, inhibiting inflammation and restoring homeostasis [38]. It is indisputable that cataract surgery is a harsh blow to ocular surface homeostasis, especially in patients with dry eyes before surgery who are already imbalanced. Thus, the aberrations of dry eye patients after cataract surgery are greater, and we detected significant aberration changes over the 4.0 mm zone after 1 month. One study reported that persistent DE symptoms after cataract surgery were associated with a high OSDI score at baseline, low TBUT, low digital pressure score and extended MG dropout at 1 month [26]. Undeniably, the surgical incision, intraoperative flushing and application of eyedrops may have short-term implications for corneal HOAs [38]. Our study also found long-term changes in posterior surface corneal HOAs after cataract surgery and agrees with findings in previous studies that these are attributed to corneal inflammation, corneal edema, keratic precipitates, incision, endothelial damage, or localized Descemet membrane detachment affecting the posterior surface more than the anterior surface [39,40]. There were no significant differences in the anterior and total aberrations between 3 months after operation and baseline. Some of these changes can be attributed to postoperative corneal remodeling.

Incision remodeling slowly continues for months to years after surgery, a process that is thought to be associated with collagen deposits on the posterior endothelial surface of the incision [41]. Late posterior wound retraction can last up to 15 years, which may imply long-term incision remodeling. The clinical effect of posterior wound retraction may induce changes in anterior and posterior corneal curvatures and thereby alter corneal power and astigmatism [42]. Cataract surgery can damage the corneal epithelium, and epithelial thickness is associated with local irregularities of corneal topography that contribute to the optical power of the cornea [43]. When the cornea is damaged, wound healing primarily progresses toward reestablishing the continuity of the corneal epithelium, followed by the remodeling of the stromal tissue via activated keratocytes [44]. Although the operation aggravated the dry eye condition, changes in corneal regularity and then increases in corneal intrinsic aberration and epithelial or stromal remodeling might restore corneal regularity to the preoperative level. There was no remission of eye dryness. Inflammation might be the internal reason for these epithelial changes. A previous study showed significant Langerhans cell infiltration at 3 days and 5 days after cataract surgery. Langerhans cell infiltration contributed to deranged epithelial renewal and the thinning of the corneal epithelium [45]. Inevitably, the infiltration of inflammatory cells will aggravate eye dryness. Thus, surgical incision leads to the direct injury and indirect injury of inflammation.

In our study, we found that the corneal vortical nerve average density and maximum length were negatively correlated with anterior corneal aberrations, including total aberrations, LOAs and HOAs. The density of the corneal subbasal nerve tends to be reduced in DED [46,47]. We wondered whether changes in the corneal nerve would lead to increased corneal intrinsic aberrations in DED. Our previous studies verified this hypothesis and considered it to have a certain impact. Only the corneal vortical nerve maximum length had a negative correlation with correlated anterior corneal aberrations after 1 month. However, these correlations disappeared after three months. The corneal vortical nerve average density and maximum length were positively correlated with posterior corneal surface aberration after 3 months. One cause may be the loss of corneal endothelium. Other corneal anterior aberration changes disappeared due to corneal remodeling. Moreover, preoperative and postoperative changes and correlations were mostly found in the corneal vortex, which proved to be a unique site related to eye dryness.

Our study has some limitations. The total corneal aberrations, corneal nerve length, corneal vortical and peripheral nerve density and length were analyzed. However, each aberration, corneal nerve width, branch number and nerve tortuosity required further analysis. The basis of the corneal remodeling is also needed, such as anterior-segment optical coherence tomography. The findings should not be generalized to cataract surgery with subthreshold preoperative DED, and follow-up research will be conducted on patients with non-dry eyes. In future research, we should distinguish the types of eye dryness for a better understanding. Longer studies should be carried out to better understand the changes in corneal nerves and aberrations after cataract surgery information about inflammatory cytokines and neurotrophins are further required to validate our findings. It must be conceded that the coefficient of determination for the corneal RMS and the corneal nerve was slightly smaller but of importance.

## 5. Conclusions

The results of this study extend our understanding of the changes in corneal intrinsic aberrations and nerve density before and after cataract surgery using a comprehensive approach. Our data show that the corneal vortical nerve maximum length and average density are decreased after cataract surgery, which may be due to the persistence of eye dryness. The corneal vortex provided a unique site to observe long-term corneal nerve injury related to eye dryness. IVCM provided an objective examination to evaluate the corneal nerve changes of patients with DED before and after cataract surgery.

## Figures and Tables

**Table 1 jcm-11-05697-t001:** Parameters of visual function, ocular symptoms and signs before and after cataract surgery.

Parameters	Baseline	1-Month	3 Months	*p* Value		
				Baseline vs. 1 Month	Baseline vs. 3 Months	3 Months vs. 1 Month
BCVA ^#^	0.59 ± 0.47	0.23 ± 0.38	0.21 ± 0.21	<0.01	<0.01	0.66
Catquest-9SF Rasch score ^$^	1.00 (−0.90, 2.71)	−0.54 (−1.82, 0.01)	−0.69 (−1.86, −0.02)	<0.01	0.03	0.57
OSDI score ^$^	37.50 (20.45, 53.73)	20.00 (9.38, 41.15)	14.29 (6.25, 25.00)	0.20	0.03	0.13
Sensitivity to light ^$^	1.00 (1.00, 3.00)	1.00 (0.00, 3.00)	1.00 (0.25, 3.00)	0.97	0.13	0.45
Foreign body sensation ^$^	1.00 (0.00, 1.00)	1.00 (0.00, 2.00)	1.00 (0.00, 1.50)	0.64	0.16	0.72
Painful eyes ^$^	0.00 (0.00, 1.00)	1.00 (0.00, 3.00)	0.50 (0.00, 2.50)	0.01	0.23	0.45
Blurred vision ^$^	1.00 (0.00, 3.00)	1.00 (0.00, 2.00)	1.00 (0.75, 2.25)	0.32	0.26	0.15
Poor vision ^$^	1.00 (1.00, 3.00)	0.00 (0.00, 1.00)	0.00 (0.00, 1.00)	<0.01	0.02	0.06
TBUT(s) ^$^	4.01 (2.87, 5.166)	3.50 (2.52, 4.91)	4.33 (3.01, 6.73)	0.54	0.43	0.72
TMH (mm) ^$^	0.16 (0.15, 0.20)	0.19 (0.15, 0.22)	0.19 (0.15, 0.23)	0.01	0.29	0.59
Meibography score ^$^	2.00 (2.00, 3.00)	3.00 (2.00, 3.00)	3.00 (2.00, 3.50)	<0.01	0.08	0.32

BCVA: best-corrected Snellen visual acuity; TBUT: tear break-up time; TMH: tear meniscus height; OSDI: ocular surface disease index; ^#^, paired *t* test; mean ± standard deviation was reported; ^$^, Wilcoxon signed rank test, median (25% quantile, 75% quantile) was reported.

**Table 2 jcm-11-05697-t002:** Comparison of the ICVM parameters before and after cataract surgery.

Parameters	Baseline	1-Month	3 Months	*p* Value		
				Baseline vs. 1 Month	Baseline vs. 3 Months	3 Months vs. 1 Month
Corneal endothelial number	2832.55 ± 344.91	2311.97 ± 559.27	2382.08 ± 585.54	<0.01	<0.01	0.08
Corneal vortical nerve maximum length (mm)	2.58 ± 0.70	2.23 ± 0.69	2.13 ± 0.85	<0.01	<0.01	0.03
Corneal vortical nerve average density (mm/mm^2^)	14.99 ± 4.11	12.98 ± 4.17	12.41 ± 5.16	<0.01	<0.01	0.01
Corneal peripheral nerve maximum length (mm)	3.02 ± 0.63	2.68 ± 0.64	2.70 ± 0.49	<0.01	<0.01	0.25
Corneal peripheral nerve average density (mm/mm^2^)	16.08 ± 3.68	13.71 ± 3.61	13.48 ± 2.95	<0.01	0.04	0.31

**Table 3 jcm-11-05697-t003:** Comparison of the total, higher-order and lower-order aberrations of the total, anterior and posterior surface of the cornea before and after cataract surgery.

Parameters	Baseline	1 Month	3 Months	*p* Value		
				Baseline vs. 1 Month	Baseline vs. 3 Months	3 Months vs. 1 Month
RMS (CF)	0.69 (0.55, 0.91)	0.75 (0.57, 0.99)	0.65 (0.45, 0.95)	0.09	0.92	0.07
RMS HOA (CF)	0.20 (0.14, 0.26)	0.23 (0.15, 0.28)	0.20 (0.13, 0.29)	<0.01	0.44	0.43
RMS LOA (CF)	0.67 (0.52, 0.87)	0.72 (0.53, 0.95)	0.62 (0.43, 0.90)	0.13	0.91	0.07
RMS (CB)	0.25 (0.21, 0.32)	0.28 (0.23, 0.35)	0.27 (0.24, 0.33)	<0.01	0.06	0.59
RMS HOA (CB)	0.07 (0.06, 0.09)	0.09 (0.07, 0.12)	0.08 (0.07, 0.11)	<0.01	0.03	0.43
RMS LOA (CB)	0.24 (0.20, 0.31)	0.26 (0.22, 0.32)	0.26 (0.22, 0.31)	<0.01	0.17	0.55
RMS (Cornea)	0.75 (0.53, 0.99)	0.83 (0.59, 1.07)	0.64 (0.45, 1.03)	0.04	0.96	0.02
RMS HOA (Cornea)	0.21 (0.15, 0.29)	0.25 (0.17, 0.33)	0.23 (0.14, 0.34)	<0.01	0.22	0.34
RMS LOA (Cornea)	0.72 (0.50, 0.95)	0.78 (0.54, 1.01)	0.62 (0.43, 0.98)	0.09	0.95	0.02
ISV	19.00 (14.00, 25.00)	21.50 (16.75, 29.00)	21.00 (15.00, 29.75)	<0.01	0.01	0.75
IVA	0.15 (0.12, 0.22)	0.19 (0.13, 0.24)	0.17 (0.11, 0.27)	0.01	0.17	0.39
IHA	4.15 (1.68, 8.05)	4.80 (2.28, 9.88)	5.20 (1.70, 9.48)	0.06	0.75	0.66
IHD	0.01 (0.01, 0.02)	0.01 (0.01, 0.02)	0.01 (0.01, 0.02)	0.18	0.89	0.01

ISV: index of surface variance; IVA: index of vertical asymmetry; IHA: index of height asymmetry; IHD: index of height decentration; RMS (CF): total root mean square (corneal front); RMS HOA (CF): root mean square of higher-order aberrations (corneal front); RMS LOA (CF): root mean square of lower-order aberrations (corneal front); RMS (CB): total root mean square (corneal back); RMS HOA (CB): root mean square of higher-order aberrations (corneal back); RMS LOA (CB): root mean square of lower-order aberrations (corneal back); RMS (cornea): total root mean square (cornea); RMS HOA (cornea): root mean square of higher-order aberrations (cornea); RMS LOA (cornea): root mean square of lower-order aberrations (cornea).

## Data Availability

The analysis data used in this study are available from the corresponding author upon request.

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
