# Peer review of "Change Patterns in Corneal Intrinsic Aberrations and Nerve Density after Cataract Surgery in Patients with Dry Eye Disease"

_jcm, 2022, doi:10.3390/jcm11195697_

Round 1

Reviewer 1 Report

 This study evaluated the change patterns in corneal aberrations and nerve density after cataract, the relusts indicated that  after cataract surgery in dry eye patients, corneal aberration changes occurred mainly on the anterior and posterior surfaces, and the corneal vortical nerve maximum length and average density are reduced after cataract surgery in patients with DED.

The new discovery in this study could add more konwledge for this field and are useful to reach efficient treatment.

1  According the the data in table 1, the dry eye state continues reduced within 3 months after surgery, but not worsen(abstract). 

2 Did the changes of corneal aberrations and verve correlated with the severity of DED?  The eye disease including diabetic related cornea changes, cornea inflammation etal should be excluded from the patients if necessary.

Author Response

Response to Reviewer 1 Comments

Dear Reviewers:

Thank you very much for your letter and for the reviewers’ comments concerning our manuscript entitled “Change patterns in corneal surface aberrations and nerve density after cataract surgery in patients with dry eye disease” (Manuscript ID: jcm-1905517). We have revised manuscript in accordance with requirements of the editors and the reviewers. We have addressed the comments raised by reviewers, and the amendments are highlighted in red in revised manuscript. Those comments are all valuable and very helpful for revising .

Reviewer 1:

This study evaluated the change patterns in corneal aberrations and nerve density after cataract, the relusts indicated that  after cataract surgery in dry eye patients, corneal aberration changes occurred mainly on the anterior and posterior surfaces, and the corneal vortical nerve maximum length and average density are reduced after cataract surgery in patients with DED.

The new discovery in this study could add more konwledge for this field and are useful to reach efficient treatment.

Point 1: According the the data in table 1, the dry eye state continues reduced within 3 months after surgery, but not worsen(abstract). 

Response 1: Thanks for your suggestion. We were really sorry for our careless mistakes. Our statement was wrong. We meant that although corneal aberration gradually recovered one month after surgery, the dry eye condition did not improve. There was no significant difference in dry eye status (OSDI score and TBUT) 1-month compared with baseline. At 3-months, the OSDI score was significantly decreased compared with baseline. However, this does not mean that the dry eye symptoms will improve 3-months after surgery. Because the visual acuity is significantly improved after cataract surgery, which affects the OSDI score. So we compared each dry eye symptom of the OSDI questionnaire. Poor vision at 3-months improved compared with baseline, other symptoms did not change significantly. 3-months after the operation, the dry eye symptoms did not improve, and the dry eye status persisted. TBUT at 3-months had no significant difference with baseline, the dry eye signs status persisted. To avoid ambiguity, we will rewrite this sentence. (Line 29-31)

Point 2: Did the changes of corneal aberrations and verve correlated with the severity of DED?  The eye disease including diabetic related cornea changes, cornea inflammation etal should be excluded from the patients if necessary.

Response 2: Thanks for your comment. We analyzed the correlation between preoperative dry eye parameters and corneal aberrations and corneal nerves, as shown in the following table.

As you can see in our manuscript (Line 83-87). Exclusion criteria include: any ocular surface and corneal abnormalities; diabetes and immunologic systemic disease. Thank you for your comments. We have refined the exclusion criteria.

TMH

TBUT

OSDI score

Sensitivity to light

Foreign body sensation

Painful eyes

Blurred vision

Poor vision

Corneal vortical nerve maximum length(mm)

NS

0.12*

NS

NS

NS

NS

NS

NS

Corneal vortical nerve average density(mm/mm2)

NS

0.13*

NS

NS

NS

NS

NS

NS

Corneal peripheral nerve maximum length(mm)

NS

0.15*

NS

0.28*

NS

NS

NS

0.33**

Corneal peripheral nerve average density(mm/mm2)

NS

0.14*

0.26*

NS

NS

NS

NS

0.28*

Corneal endothelial number

NS

NS

NS

NS

NS

NS

NS

NS

RMS (CF)

NS

NS

NS

NS

NS

NS

NS

NS

RMS HOA (CF)

NS

-0.15*

NS

NS

NS

NS

NS

NS

RMS LOA (CF)

NS

NS

NS

NS

NS

NS

NS

NS

RMS (CB)

NS

NS

0.20*

NS

NS

NS

NS

NS

RMS HOA (CB)

NS

NS

NS

NS

NS

NS

NS

NS

RMS LOA (CB)

NS

NS

0.20*

NS

NS

NS

NS

NS

RMS (Cornea)

NS

NS

NS

NS

NS

NS

0.20*

NS

RMS HOA (Cornea)

NS

-0.16*

NS

NS

NS

NS

NS

NS

RMS LOA (Cornea)

NS

NS

NS

NS

NS

NS

0.21*

No

We tried our best to improve the manuscript and made some changes in the manuscript. These changes will not influence the content and framework of the paper. And here we did not list the changes but marked in red in revised paper.

We appreciate for your warm work earnestly, and hope that the correction will meet with approval.

Once again, thank you very much for your comments and suggestions.

Reviewer 2 Report

This manuscript described a study using aberrometry and IVCM to detect

corneal surface aberrations and nerve density after cataract surgery

in patients with dry eye disease. Although I do not have special

comments on this manuscript, can the authors reply the following

details?

  1. Was the corneal estheiometry performed before and/or after surgery?

  2. Did the authors exclude cases of neurothrophic ocular diseases

such as previous viral infectious keratitis?

  3. Besides total HOAs, were other HOAs such as coma, trefoil or

spherical aberration individually examined?

  4. What were the pre- and posterative medications given to the

patients in this study?

Author Response

Response to Reviewer 2 Comments

Dear  Reviewers:

Thank you very much for your letter and for the reviewers’ comments concerning our manuscript entitled “Change patterns in corneal surface aberrations and nerve density after cataract surgery in patients with dry eye disease” (Manuscript ID: jcm-1905517). We have revised manuscript in accordance with requirements of the editors and the reviewers. We have addressed the comments raised by reviewers, and the amendments are highlighted in red in revised manuscript. Those comments are all valuable and very helpful for revising .

Reviewer 2:

This manuscript described a study using aberrometry and IVCM to detect corneal surface aberrations and nerve density after cataract surgery in patients with dry eye disease. Although I do not have special comments on this manuscript, can the authors reply the following details?

Point 1: Was the corneal estheiometry performed before and/or after surgery?

Response 1: Thank you for the comments. We agree that measuring the function changes of the altered corneal nerves would enrich the result of the research. We reviewed previous publications in regards to corneal nerve changes and found that a reliable association between corneal nerve morphology and its sensory function has not been established. So, given the fact that esthesiometry measuring could be affected by multiple factors, we decided to focus on morphological alterations of corneal nerves and did not involve esthesiometry in this research. We would perform esthesiometry in further studies.

Point 2: Did the authors exclude cases of neurothrophic ocular diseases such as previous viral infectious keratitis?

Response 2: Thank you for the comments. As you can see in our manuscript (Line 83-87). Exclusion criteria include: any ocular surface and corneal abnormalities; diabetes and immunologic systemic disease. Thank you for your comments. We have refined the exclusion criteria.

Point 3: Besides total HOAs, were other HOAs such as coma, trefoil or spherical aberration individually examined?

Response 3: Thank you for the comments. Our findings include:including the total root mean square (RMS) aberrations, RMS of the higher-order aberrations (RMS HOA), and RMS of the lower-order aberrations (RMS LOA). The integrated software of Pantacam automatically generated the Zernike polynomials of the cornea and up to the 8th Zernike co-efficients. See the table below for details. It contains other HOAs such as coma, trefoil or spherical aberration,  However, our subject is to preliminarily explore the changes of higher-order aberrations, so we do not analyze other higher-order aberrations separately. In future research, we will consider including higher-order aberrations at all levels.

Point 4: What were the pre- and posterative medications given to the patients in this study?

Response 4: Thank you for the comments. This is very important for us. We need to describe the preoperative and postoperative medication of patients in detail in the text. (Line 82-83, Line 138-140, Line 146-148). Patients were not given any eye drops except artificial tears before enrollment. All patients were given levofloxacin 0.5% eye drops (Santen, Japan) four times a day from three days before surgery. Prednisolone acetate 1% eye drops (Allergan, Ireland) and levofloxacin 0.5% eye drops were used four times per day for one week and were then gradually reduced in both groups.

We tried our best to improve the manuscript and made some changes in the manuscript. These changes will not influence the content and framework of the paper. And here we did not list the changes but marked in red in revised paper.

We appreciate for your warm work earnestly, and hope that the correction will meet with approval.

Once again, thank you very much for your comments and suggestions.

Reviewer 3 Report

The authors evaluate the change patterns in corneal aberration and nerve density cataract surgery in DED and they found that corneal aberration occurred mainly on the anterior and posterior surface. The maximum length and average density of corneal nerve are reduced after cataract surgery.

Title

If the corneal anterior and posterior is examined why the title is corneal surface?

Only the anterior surface was measured?

Described the type of DED

Abstract

Remove headings

Limit is 200 words

Include number results in the results section

The conclusion is completely misleading and need to rewrite to improve the comprehension

Introduction

The novelty of the research is poor since is the mix of previous studies

Physiopathology of the reduced in corneal nerve in DED and cataract should be included in this section

Method

Divide in standard section, design, ethic, patient, procedure, materials, statistical analysis.

Line 74 – sample size in the statistical section please

Described the type of DED

Patients have medication?

Results

P value with value 0.00 does not exist rewrite with coherence of statistics

Format of the tables should be rewritten

In the table 4 if not significant correlation was achieved do noy described this big table to only the significant, described only the clinical and statistically significant and in discussion give the rationale of the correlation.

The 210 – 215 discussion is very vague a do not give a crucial reason, improve the explanation of MGD link

220 – the topic medication could influence TMH, you discussion lines are somewhat risky

The discussion is quite redundant and do not give the reason clearly, if your results could not support your discussion lines please avoid or remove

Include only the information that you demonstrated with your findings

The limitation and future research section should be improved and extended due the limitations of the study

The conclusion should be rewrite in order to improve the comprehension, it is wordy redacted.

In the references list, when possible, please include only references after 2010 and in addition when possible include only references included in the Journal Citation Reports or indexed and reputed Scientific Journals.

Author Response

Response to Reviewer 3 Comments

Dear  Reviewers:

Thank you very much for your letter and for the reviewers’ comments concerning our manuscript entitled “Change patterns in corneal intrinsic aberrations and nerve density after cataract surgery in patients with dry eye disease” (Manuscript ID: jcm-1905517). We have revised manuscript in accordance with requirements of the editors and the reviewers. We have addressed the comments raised by reviewers, and the amendments are highlighted in red in revised manuscript. Those comments are all valuable and very helpful for revising .

Reviewer 3:

The authors evaluate the change patterns in corneal aberration and nerve density cataract surgery in DED and they found that corneal aberration occurred mainly on the anterior and posterior surface. The maximum length and average density of corneal nerve are reduced after cataract surgery.

Point 1: Title

1)If the corneal anterior and posterior is examined why the title is corneal surface?

Only the anterior surface was measured?

2)Described the type of DED

Response 1: 1)Thank you for the comments. Your opinion is very critical. We have revised the title of the article to make it more in line with our main purpose. We collected data including the corneal aberrations which comprised the total root mean square (RMS) aberrations, RMS of the higher-order aberrations (RMS HOA), and RMS of the lower-order aberrations (RMS LOA) for the anterior, posterior, and total corneal surfaces in a 4 mm pupil setting.

2)Besides, patients were enrolled in the research according to the diagnostic criteria for DED established by the Tear Film and Ocular Surface Society (TFOS) Dry Eye Workshop II in 2017. We did not distinguish aqueous and evaporative dry eye cases in the present research. We will further investigate the different DED types in future.

Point 2: Abstract

Remove headings

Limit is 200 words

Include number results in the results section

The conclusion is completely misleading and need to rewrite to improve the comprehension

Response 2: Thank you for the comments. We have modified the structure of the abstract to meet the requirements of the magazine. We have revised the results and conclusions to make it better. (Line 19-32)

Point 3: Introduction

1)The novelty of the research is poor since is the mix of previous studies

2)Physiopathology of the reduced in corneal nerve in DED and cataract should be included in this section

Response 3: 1)Thanks for the comments. Previous studies did not pay attention to corneal intrinsic aberrations. There are many similarities in ocular surface changes between the two. Prior studies have focused on one specific aspect: changes in corneal nerves or aberrations. In our study, we using Pentacam to measure the corneal intrinsic aberrations before and after cataract surgery. Besides, we selected each of 5 high-quality images in the corneal vortex and periphery with no overlapping for analysis of the corneal nerve parameters by CNS-Net. And we found that there was a significant decrease in the corneal vortical nerve maximum length and average density after the operation, and this damage was present after 3-months. We thanks for your insightful comments. In the future, we will strive to improve novelty in further research.

2)According to your opinion, we carefully read the relevant references and added physiopathology of the reduced in corneal nerve in DED and cataract in the introduction section. (Line 44-51)

Point 4: Method

1) Divide in standard section, design, ethic, patient, procedure, materials, statistical analysis.

2)Line 74 – sample size in the statistical section please

3)Described the type of DED

4)Patients have medication?

Response 4: 1)Thank you for the comments. We have modified the structure of the method to meet the requirements of the magazine. 

2)Sample size calculation part was put into statistical analysis section. (Line 150-152)

3)For type of DED, patients were enrolled in the research according to the diagnostic criteria for DED established by the Tear Film and Ocular Surface Society (TFOS) Dry Eye Workshop II in 2017. We did not distinguish aqueous and evaporative dry eye cases in the present research. We will further investigate the different DED types in future.

4)Patients medication: We need to describe the preoperative and postoperative medication of patients in detail in the text. (Line 82-83, Line 138-140, Line 146-148). Patients were not given any eye drops except artificial tears before enrollment. All patients were given levofloxacin 0.5% eye drops (Santen, Japan) four times a day from three days before surgery. Prednisolone acetate 1% eye drops (Allergan, Ireland) and levofloxacin 0.5% eye drops were used four times per day for one week and were then gradually reduced in both groups.

Point 5: Results

1)P value with value 0.00 does not exist rewrite with coherence of statistics

2)Format of the tables should be rewritten

3)In the table 4 if not significant correlation was achieved do noy described this big table to only the significant, described only the clinical and statistically significant and in discussion give the rationale of the correlation.

 Response 5: 1)Thank you for the comments .This is our fault. Our P = 0.00 means P < 0.01, because we only keep two decimal places after the decimal point. We have made corresponding modifications.

2)We have modified the table to an editable version.

3)Based on your comments, we deleted Table 4 and described the useful contents in detail in the article.

Point 6: 1)The 210 – 215 discussion is very vague a do not give a crucial reason, improve the explanation of MGD link 

2)220 – the topic medication could influence TMH, you discussion lines are somewhat risky

3)The discussion is quite redundant and do not give the reason clearly, if your results could not support your discussion lines please avoid or remove

Include only the information that you demonstrated with your findings

4)The limitation and future research section should be improved and extended due the limitations of the study

 Response 6: 1)Thank you for the comments. We rewrote this part. (Line 244-252)

  • Thank you for the comments. We rewrote this part. (Line 258-263)
  • We feel great thanks for your professional review work on our article. We carefully read the discussion section and made a comprehensive revision. We deleted the redundantpart of the discussion.
  • Thanks for the comments. We improved and extended due the limitations of the study(Line 356-359)

Point 7: The conclusion should be rewrite in order to improve the comprehension, it is wordy redacted.

Response 7: Thank you for the comments. We rewrote the conclusion section.

Point 8: In the references list, when possible, please include only references after 2010 and in addition when possible include only references included in the Journal Citation Reports or indexed and reputed Scientific Journal

Response 8: Thank you for the comments. We revised the references and changed the reference year to after 2010.

We tried our best to improve the manuscript and made some changes in the manuscript. These changes will not influence the content and framework of the paper. And here we did not list the changes but marked in red in revised paper.

We appreciate for your warm work earnestly, and hope that the correction will meet with approval.

Once again, thank you very much for your comments and suggestions.

Reviewer 4 Report

The authors have accumulated many data.

As it is  a sensitive matter that addresses  many patients  and has  potentially a significant  impact on the subject the wording should be very diligent when outlining the problem.  There is a difference to say that ” There are numerous causes of exacerbation”  or  ”Severeal  causes of potential exacerbation …..  have been suggested”. It is the  tendency  of jumping over the lack  of  proof of  causal relation ship in some  statements that easily creates the missconception thata causal relation ship does exist , where, instead, it does not. 

The same  applies considering the conclusions:  The authors claim that   the  data show that corneal vortical nerve   maximum length and average density are decreased after cataract surgery partly because  of the aggravation of dry eye conditions.

That  is not correct as the methods do not allow such a conclusion because the study design does not allow  the establishement of  such causal realtionship.  The results presented  show that  that corneal vortical nerve length and average density are decreased after cataract surgery,   However, an  aggaravtion of dry eye complaint  did apparently  not occur. In the contrary as  the OSDI score decreased  from   37,5   over  20,0   ( 1 month)  to 14,3  ( 3 months there seems to be an improvement.    The foreign body sensation did not show any significant difference, similarly as  TBUT   did not alter significantly  ( table 1) .  As it presumed that, as usual, a  A p -value less than 0.05 (typically ≤ 0.05) is statistically significant..

Accordingly also the results as reflected in table 4  not show a convinging significance.

As to the methods and results,some issues attract attention:

Here, a siginifcant deficit  is  the missing information as  to the exact location opf the  incision which matters as to  postop refraction   and measurements.

Further, in the methods  section, it is unclear , if baseline means  the same as before cataract surgery.

 As to results:

The values of table 3  comparing p values are not conclusive as  in several parameters P  values as provided suggests  significant difference  between  baseline and 1 month with  the loss of  such significance  between baseline  and 3 months for the majority of parameters, suggesting normalization. 

In the lay out reference numbers have double  numbers as identifiers which needs to be corrected.

Author Response

Response to Reviewer 4 Comments

Dear  Reviewers:

Thank you very much for your letter and for the reviewers’ comments concerning our manuscript entitled “Change patterns in corneal surface aberrations and nerve density after cataract surgery in patients with dry eye disease” (Manuscript ID: jcm-1905517). We have revised manuscript in accordance with requirements of the editors and the reviewers. We have addressed the comments raised by reviewers, and the amendments are highlighted in red in revised manuscript. Those comments are all valuable and very helpful for revising .

Reviewer 4:

The authors evaluate the change patterns in corneal aberration and nerve density cataract surgery in DED and they found that corneal aberration occurred mainly on the anterior and posterior surface. The maximum length and average density of corneal nerve are reduced after cataract surgery.

Point 1: 1)As it is  a sensitive matter that addresses  many patients  and has  potentially a significant  impact on the subject the wording should be very diligent when outlining the problem.  There is a difference to say that ” There are numerous causes of exacerbation”  or  ”Severeal  causes of potential exacerbation …..  have been suggested”. It is the  tendency  of jumping over the lack  of  proof of  causal relation ship in some  statements that easily creates the missconception thata causal relation ship does exist , where, instead, it does not. 

2)The same  applies considering the conclusions:  The authors claim that   the  data show that corneal vortical nerve   maximum length and average density are decreased after cataract surgery partly because  of the aggravation of dry eye conditions.

That  is not correct as the methods do not allow such a conclusion because the study design does not allow  the establishement of  such causal realtionship.  The results presented  show that  that corneal vortical nerve length and average density are decreased after cataract surgery,   However, an  aggaravtion of dry eye complaint  did apparently  not occur. In the contrary as  the OSDI score decreased  from   37,5   over  20,0   ( 1 month)  to 14,3  ( 3 months there seems to be an improvement.    The foreign body sensation did not show any significant difference, similarly as  TBUT   did not alter significantly  ( table 1) .  As it presumed that, as usual, a  A p -value less than 0.05 (typically ≤ 0.05) is statistically significant..

3)Accordingly also the results as reflected in table 4  not show a convinging significance.

Response 1: 1)Thanks for the comments. Your opinions are very important to us. Based on your opinions, we have read the whole in detail, adjusted the wording and the expression of the results to make them more reasonable and understandable.

  • We rewrote the conclusion to make it more reasonable.

3)we deleted Table 4 and described the useful contents in detail in the article. Besides, We readjusted the discussion section to correspond to the results.

Point 2:As to the methods and results,some issues attract attention:

1)Here, a siginifcant deficit  is  the missing information as  to the exact location opf the  incision which matters as to  postop refraction   and measurements.

2)Further, in the methods  section, it is unclear , if baseline means  the same as before cataract surgery.

Response 2: 1)Thanks for the comments. As you said, the position of corneal incision is very important for refractive change after surgery, which we have added in the article. (Line 142-143) 2)Baseline means  the same as before cataract surgery, we will explain it in the method. (Line 96)

Point 3:As to results:

The values of table 3  comparing p values are not conclusive as  in several parameters P  values as provided suggests  significant difference  between  baseline and 1 month with  the loss of  such significance  between baseline  and 3 months for the majority of parameters, suggesting normalization. 

Response 3: Thanks for the comments. We readjusted the discussion section to correspond to the results. What you said is quite right. Patients had increased anterior and total corneal HOAs at 1 month. Significant changes in posterior corneal HOAs occurred at 1 month and 3 months. Therefore, at three months after surgery, only the high-order aberration of the posterior surface of the cornea was still significantly different from that before surgery.

Point 4: In the lay out reference numbers have double  numbers as identifiers which needs to be corrected.

Response 4: Thanks for your comments. We revised the numbers in the references.

We tried our best to improve the manuscript and made some changes in the manuscript. These changes will not influence the content and framework of the paper. And here we did not list the changes but marked in red in revised paper.

We appreciate for your warm work earnestly, and hope that the correction will meet with approval.

Once again, thank you very much for your comments and suggestions.

Reviewer 5 Report

Table 4 presents a comparison of multiple correlations and their significance level. Was any statistical system used to counteract the multiple comparisons such as the Bonferroni correction?  

Line 175 "the corneal endothelial number was negatively associated with posterior corneal surface aberrations." This does not seem to correlate with the data presented in table 4 where the corneal endothelial number was only negatively correlated with RMS HOA of the front cornea.

In lines 199 and 215 "Oculus" is mentioned instead of "Oculus keratograph"

Author Response

Response to Reviewer 5 Comments

Dear  Reviewers:

Thank you very much for your letter and for the reviewers’ comments concerning our manuscript entitled “Change patterns in corneal surface aberrations and nerve density after cataract surgery in patients with dry eye disease” (Manuscript ID: jcm-1905517). We have revised manuscript in accordance with requirements of the editors and the reviewers. We have addressed the comments raised by reviewers, and the amendments are highlighted in red in revised manuscript. Those comments are all valuable and very helpful for revising .

Reviewer 5:

Point 1: Table 4 presents a comparison of multiple correlations and their significance level. Was any statistical system used to counteract the multiple comparisons such as the Bonferroni correction?  

Response 1: Thanks for the comments. The correlation comparison in Table 4 was bi-variable correlation, not multiple correlation or multiple comparison. So we did not the Bonferroni correction. Table 4 is very lengthy, and many results are not significantly related. We deleted Table 4 and described only the clinical and statistically significant in result section .

Point 2: Line 175 "the corneal endothelial number was negatively associated with posterior corneal surface aberrations." This does not seem to correlate with the data presented in table 4 where the corneal endothelial number was only negatively correlated with RMS HOA of the front cornea.

Response 2: Thanks for your suggestion. We were really sorry for our careless mistakes. We made corresponding modifications. (Line 192-194)

Point 3: In lines 199 and 215 "Oculus" is mentioned instead of "Oculus keratograph"

Response 3: Thanks for the comments. This is our mistake, we have replaced "Oculus" with "Oculus keratograph".(Line 233,258)

We tried our best to improve the manuscript and made some changes in the manuscript. These changes will not influence the content and framework of the paper. And here we did not list the changes but marked in red in revised paper.

We appreciate for your warm work earnestly, and hope that the correction will meet with approval.

Once again, thank you very much for your comments and suggestions.

Round 2

Reviewer 3 Report

The authors solved the comment proposed by the reviewer

Thanks for the revision